# Tomato Biodiversity and Drought Tolerance: A Multilevel Review

**DOI:** 10.3390/ijms241210044

**Published:** 2023-06-12

**Authors:** Veronica Conti, Luigi Parrotta, Marco Romi, Stefano Del Duca, Giampiero Cai

**Affiliations:** 1Department of Biological, Geological and Environmental Sciences, University of Bologna, 40126 Bologna, Italy; veronica.conti8@unibo.it (V.C.); luigi.parrotta@unibo.it (L.P.); 2Department of Life Sciences, University of Siena, 53100 Siena, Italy; marco.romi@unisi.it (M.R.); giampiero.cai@unisi.it (G.C.); 3Interdepartmental Center for Agri-Food Industrial Research, University of Bologna, 40126 Bologna, Italy

**Keywords:** tomato, drought stress, physiological responses, biochemical responses, genetic features, fruit quality

## Abstract

Ongoing global climate change suggests that crops will be exposed to environmental stresses that may affect their productivity, leading to possible global food shortages. Among these stresses, drought is the most important contributor to yield loss in global agriculture. Drought stress negatively affects various physiological, genetic, biochemical, and morphological characteristics of plants. Drought also causes pollen sterility and affects flower development, resulting in reduced seed production and fruit quality. Tomato (*Solanum lycopersicum* L.) is one of the most economically important crops in different parts of the world, including the Mediterranean region, and it is known that drought limits crop productivity, with economic consequences. Many different tomato cultivars are currently cultivated, and they differ in terms of genetic, biochemical, and physiological traits; as such, they represent a reservoir of potential candidates for coping with drought stress. This review aims to summarize the contribution of specific physio-molecular traits to drought tolerance and how they vary among tomato cultivars. At the genetic and proteomic level, genes encoding osmotins, dehydrins, aquaporins, and MAP kinases seem to improve the drought tolerance of tomato varieties. Genes encoding ROS-scavenging enzymes and chaperone proteins are also critical. In addition, proteins involved in sucrose and CO_2_ metabolism may increase tolerance. At the physiological level, plants improve drought tolerance by adjusting photosynthesis, modulating ABA, and pigment levels, and altering sugar metabolism. As a result, we underline that drought tolerance depends on the interaction of several mechanisms operating at different levels. Therefore, the selection of drought-tolerant cultivars must consider all these characteristics. In addition, we underline that cultivars may exhibit distinct, albeit overlapping, multilevel responses that allow differentiation of individual cultivars. Consequently, this review highlights the importance of tomato biodiversity for an efficient response to drought and for preserving fruit quality levels.

## 1. Introduction

The average temperature on Earth is constantly rising due to global warming. Since the Industrial Revolution, global average temperatures have risen by 1.1 °C. The Intergovernmental Panel on Climate Change (IPCC) warns of widespread, rapid, and intensifying climate change effects and provides new estimates of the likelihood of exceeding the 1.5 °C or 2 °C global warming threshold in the coming decades unless immediate, rapid, and large-scale reductions in greenhouse gas emissions are implemented. Average global temperatures will rise by 2.1–3.5 °C in a scenario with little change from current global-development patterns [1].

Severe weather events, such as droughts, will become more common as average temperatures rise, and the frequency and severity of such occurrences may keep increasing in most geographical regions in the future [1]. Global warming is also responsible for reduced precipitation in high-risk areas, such as the Mediterranean, which is predicted to become a “hot zone” in the 21st century. It is unlikely, although desirable, that the world will experience a reversal of climate change, especially in the coming years [2]. Therefore, there is an urgent need to assess the impacts of water scarcity on natural and anthropogenic ecosystems. Water demand for agriculture could double by 2050, while freshwater availability is expected to decrease by 50% due to climate change [2]. Given that modern agriculture uses large amounts of water, the impact of water scarcity on crop productivity and associated costs should not be underestimated [3]. In the past decade, global crop production losses due to drought, amounted to approximately $30 billion. The world’s population has grown from 5 billion in 1990 to more than 7.5 billion today and is predicted to reach about 10 billion by 2050. It is estimated that 5 billion people will be living in water-stressed regions by that time [4].

Numerous studies have shown that both heat and drought stress have negative effects on plant growth and physiology [5,6]. High temperature stress can cause a variety of physiological, biochemical, and molecular responses, including stomatal closure due to low vapor pressure, limiting the supply of carbon dioxide (CO_2_) [7,8]. This can also impair the photosynthetic apparatus and decrease photosystem activity, resulting in a decrease in the photosynthetic rate (Fv/Fm) and associated physiological responses, such as decreased chlorophyll content and increased proline concentration [9,10]. As a result, high temperatures frequently have a negative impact on plant growth and development [11]. Meanwhile, drought stress can cause cell turgor loss and decreased water content [12], limiting growth and dry mass accumulation [13]. To counteract the negative effects of stress conditions, plants frequently close their stomata to limit water loss, but at the expense of photosynthesis rate [14]. High temperatures and drought are frequently associated with climate change [15]. Extremely high temperatures can cause rapid soil evaporation and increase plant transpiration by inducing stomatal opening, exacerbating drought stress [16]. High temperatures and drought, when combined, were found to reduce the photosynthetic rate and PSII photochemical efficiency more than individual stresses [17].

Plants have evolved in nature to cope with drought stress through a series of morphological, physiological, and biochemical adaptations based on the high diversity of species grown in climatic regions with extreme drought conditions [18]. Plants also have strategies to prevent water loss, balance optimal water supply to critical organs, maintain cellular water content, and survive drought [3]. The ability of plants to perceive signals of water shortage and to initiate coping strategies in response is referred to as “drought resistance”. Drought resistance is a complex trait that operates through several mechanisms: (i) escape (accelerating the plant’s reproductive phase before stress impairs its survival), (ii) avoidance (increasing internal water content and preventing tissue damage), and (iii) tolerance (resisting low internal water content while maintaining growth during drought) [18].

Native to the Andean region of South America, the tomato (*Solanum lycopersicum* L.) is the second most cultivated vegetable in the world after the potato, with about 189 million tons grown on 5.16 Mha, according to the Food and Agriculture Organization of the United Nations (FAOSTAT, 2023; http://www.fao.org/faostat/en/#data/QC; (accessed on 16 March 2023)). Tomatoes were imported to Europe in 1540, but widespread cultivation did not begin until the second half of the 17th century. Tomatoes are mainly popular as a food and, because of this, are also important economically, especially in the agricultural sector. The following are some of the economic benefits that tomato plants provide: (a) In the food industry, tomatoes are used in a wide variety of foods, including sauces, soups, juices, and ketchup. The demand for these products has fueled the growth of the tomato processing industry, which employs thousands of people and generates billions of dollars in revenue each year. (b) In agriculture, tomato crops are widely grown in temperate and tropical regions, providing employment for farmers and farm workers. Global tomato production is estimated at approximately 170 million tons per year, making it one of the world’s most important food crops. (c) In exports, tomato plants are exported to a variety of countries, helping to maintain trade balances and boosting the economies of the producing countries. China, the United States, Turkey, and Italy are among the major tomato producing countries. (d) In retail, tomatoes are widely available in supermarkets, and their popularity as a staple food has fueled retail growth. Each year, retail sales of fresh tomatoes generate billions of dollars in revenue. Not to mention that tomatoes represent an important source of food and money for many people, especially in developing countries [19], as well as a rich source of health-promoting compounds, such as vitamins, carotenoids, and phenolic compounds [20].

Tomatoes have been cultivated in Europe for about 400 years, with breeding activities carried out in the last eight decades. To date, more than 10,000 tomato cultivars have been developed [21]. Intensive breeding activities conducted by scientists and breeders on the single species *S. lycopersicum* at the turn of the 20th century resulted in a wide range of morphologically distinct cultivars with large variations in fruit weight, fruit size and shape, and color. Modern tomato breeding initiatives for food market usage have traditionally focused on stress tolerance, yield, and quality features, such as firmness, color, texture, and fruit appearance traits rather than on long-term production and nutritional properties [21]. Tomato is a member of the Solanaceae family, which includes over 3000 species from both the Old and New Worlds (eggplant in China and India and pepper/potato/tomato in Central and South America). The Solanaceae phylogeny has recently been revised, and the genus *Lycopersicon* has been reintegrated into the genus *Solanum* under the new nomenclature. The cultivated tomato (*S. lycopersicum*) and 12 additional wild relatives are all members of the *Solanum* section *Lycopersicon*. The only domesticated species is *Solanum lycopersicum* [22].

Tomato production faces a number of challenges worldwide, including high input costs (seeds, fertilizers, pesticides, and irrigation), pests and diseases caused by nematodes, viruses, fungi, and bacteria (that can reduce yield and quality), and postharvest losses due to inefficient handling, storage, transportation, and processing [19]. The development of tomato cultivars with enhanced abiotic stress tolerance is one of the most sustainable approaches for its successful production. In this regard, efforts are being made to understand the mechanism of stress tolerance, gene discovery, and the interaction of genetic and environmental factors. Several -omics approaches, tools, and resources have already been developed for tomato breeding; in fact, modern sequencing technologies have greatly accelerated genomics and transcriptomics studies in tomato. These advances facilitate quantitative trait loci (QTL) mapping, genome-wide association studies (GWAS), and genomic selection (GS). It follows that under stress conditions, tomatoes self-regulate to adapt to the existing stresses by controlling gene expression, protein synthesis, and metabolite production, implying that it is essential to elucidate the functions of newly identified stress-responsive genes to understand the abiotic stress responses of plants [23].

A number of phenotyping platforms have been explored to characterize the drought stress response of plants. These include the use of optical sensors to monitor plant photosynthetic activity, growth status, and total water content. While much of the focus has been on the aerial part of the plant, the importance of phenotyping the root system has also been recognized. However, continuous real-time monitoring remains the exception rather than the rule, and most often the physiological status of the plant is measured indirectly. A recent development has been a graphene sensor able to monitor in real time the transport of water from plant roots to leaves [24], while an integrated electrochemical chip-on-plant has been used to detect gene expression under stress conditions in tobacco leaves [25].

This review will focus on the different molecular mechanisms underlying a specific challenge faced by tomato plants, namely drought stress and tolerance [26]. The different mechanisms will be compared, considering that tomato cultivars differ. The study of the diversity of responses at the variety level will shed light on the complexity of tomato drought stress responses. Thus, the core of this review is a critical assessment of the importance of conserving and valorizing tomato biodiversity as a gene pool against abiotic stress conditions, especially in the light of ongoing climate change. This review is divided into sections covering gene expression, biochemistry, metabolism, and physiology. We also attempted to emphasize how drought response increases fruit nutritional content, which can improve fruit quality. Since the availability of specific tomato varieties is sometimes limited (as is their large-scale cultivation), the study of specific tomato varieties, such as ‘Micro-Tom’ (selected for its ease of cultivation, adaptability to laboratory conditions, and short life cycle) should not be underestimated, as it could allow the achievement of rapid results. Tomato is one of the most widely cultivated crops in the world, and the literature on the effects of environmental changes is extensive. Herein, we emphasize (1) the importance of studying a specific environmental constraint (drought), which is known to be the most critical stress for tomato plants, and (2) the multifaceted responses of tomato to cope with the effects of drought. We highlight the importance of studying the different tomato cultivars with the aim of uncovering the diverse and multiple responses to drought that cultivars may have independently evolved. This study put emphasis on the conservation and valorization of the pool of tomato genotypes, increasing the concept of tomato biodiversity and providing bases to breeding programs for still productive plants in the scenario of climate change. Since tolerance is likely to be allocated to different levels, we chose to assess the importance of three different aspects (genetic, biochemical, and physiological) and their relative contribution to drought tolerance. First, we analyzed the tomato genes involved in responses to stress, then we highlighted the protective role of proteins. Next, the physiological changes and adaptations of the tomato were reported, and finally, we focused on the fruit quality levels that may be affected by drought stress.

## 2. Gene-Based Resistance to Drought

Tomato plants can adapt to drought in a variety of ways, from changes in gene expression to alterations in their physiomorphological characteristics. Responses may take the form of biochemical adaptations, mainly hormone levels, but may also involve changes in osmolyte content; for example, proline has been found to accumulate in the leaves of tolerant varieties and to have a positive effect on transpiration and leaf water potential. This is the case for long-storage tomatoes, which are particularly resistant to water shortage [27]. Responses to drought also occur as physiological changes, such as stomatal conductance, photosynthetic pigment levels, changes in water distribution and storage, and mechanical responses, such as stomatal closure. In addition, adaptation can result in changes in mesophyll structure, stomatal size, and density. Not all of these changes occur in all tomato cultivars, but often they are specific and characterize a particular cultivar. Examples of drought-tolerant tomato varieties include Ramellet, which is drought-tolerant due to its long-term adaptations [28] and the drought-tolerant cultivar ‘Tomàtiga de Ramellet’; the latter has a more compact parenchyma, a large number of xylem vessels, wider phloem vessels, and thinner stomata [29]. The number and spacing of stomata may also differ between tolerant and susceptible cultivars [30]. Changes at the biochemical, physiological, and morphological levels are all dependent on changes at the genomic and gene level. Therefore, we have chosen to organize this review starting at the gene level and discussing the role of specific genes in drought tolerance.

Since 2012, the entire tomato genome has been sequenced [31], revealing a number of promising genes for abiotic stress resistance. The tomato genome has uncovered several genes related to abiotic stress resistance, such as salt and drought tolerance, which can be used to increase tomato yields under adverse conditions. The study of gene expression allowed the identification of many drought-responsive genes and non-coding RNAs [32]. RNA-seq analysis of tomato plants under ABA treatment revealed the influence of this hormone on the expression of stress-responsive genes [33]. Furthermore, studies have revealed the regulatory role of the SlbZIP1 gene, as well as the importance of genes for endochitinase, peroxidases, and lipid transferase proteins [32]. By transcriptome analysis, 966 differentially expressed genes (DEGs) were identified under drought stress, including genes for heat shock proteins, cell wall-related enzymes and histones [34], as well as genes involved in amino acid metabolism, ethylene, and jasmonic acid signaling [35]. Overexpression of the MYB49 transcription factor was found to improve drought tolerance by reducing ROS accumulation [36], while 7 out of 99 zinc-finger transcription factor genes were found to be differentially expressed under water deficit [37].

In addition to gene expression analysis, single nucleotide polymorphisms (SNPs) and simple sequence repeats (SSRs) used in genotyping techniques could be exploited to develop genetic markers [38,39]. These markers could then be used to identify tomato varieties with higher abiotic stress resistance. Molecular markers have also contributed to the progressive identification and confirmation of local (autochthonous) tomato varieties or cultivars, which have rapidly become important as genetic resources for crop improvement due to their naturally enhanced tolerance to abiotic stresses [40,41].

Although screening different tomato varieties for improved abiotic stress tolerance using SSR markers is a promising tool for identifying new, more tolerant genotypes, it is a time-consuming process. To speed up the process, molecular phenotyping techniques can be used to assess the potential for stress tolerance in different tomato cultivars. Alternatively, genetic transformation is a faster method for producing drought-tolerant varieties than breeding. There are numerous examples in the literature of transformation by overexpression of specific genes leading to physiological or metabolomic improvements and the establishment of abiotic stress-resistant genetic traits in tomato plants [42,43]. In addition, genetic transformation is advantageous over breeding because it can be used to introduce abiotic stress tolerance genes that may not be present in wild or cultivated tomato varieties or cultivars. *MdEPF2 (Malus domestica* epidermal patterning factor) is a functional ortholog of *EPF2* in *Arabidopsis* and can be used to improve drought tolerance and water use efficiency (WUE) in crops. *EPF* from apple was ectopically expressed in *S. lycopersicum* cultivar ‘Micro-Tom’, and transgenic plants showed higher values for relative leaf water content, chlorophyll, photosynthetic rates (Fv/Fm), and WUE than wild type (WT). Overexpression of *MdEPF* regulates stomatal development, and the expression of this gene was also significantly induced by application of ABA [44]. In some cases, tolerant mutants may have higher photosynthetic efficiency (as measured by Fv/Fm) and stomatal conductance; this is the case of the overexpression of *cwInv* (cell wall invertase) [45], *AtGAMT1* [31], or *SlADL1* [46], which are the biosynthetic genes for pipecolic acid (Pip). Other examples include the *A. thaliana* gene *ATHB-7*; when overexpressed in tomato, it reduces stomatal density and makes the plant more stress tolerant [47]. Overexpression of *SlPIP2;1*, *SlPIP2;7*, and *SlPIP2;5* promotes aquaporin synthesis, increased water content, and maintenance of osmotic balance [48]. By contrast, overexpression of the osmotin gene results in increased leaf expansion, higher chlorophyll and proline content, and better maintenance of high relative water content (RWC) [49]. Osmotin may also be involved in the compartmentalisation of solutes, the protection of the native structure of proteins, or the repair of their denaturation [49]. Overexpression of the dehydrin gene (*TAS14*) reduces osmotic potential while increasing solute (sugar and K^+^) and abscisic acid (ABA) content, resulting in improved plant stress tolerance, as indicated by shoot and fruit biomass [50]. *SlMAPK3* has multiple roles in drought tolerance, including increased photosynthetic activity and osmoprotection [51,52]. Other genes are involved in maintaining tolerance to drought stress by counteracting oxidative stress. The *SlJUB1* gene was discovered in *S. lycopersicum* L. and was found to increase stress tolerance by enhancing RWC and reducing H_2_O_2_. It also activates the DELLA genes, which are involved in growth repression, supporting further drought stress resistance [53,54,55]. The cultivar with high anthocyanin content results in greater tolerance to drought by stimulating the expression of genes involved in proline biosynthesis and in superoxide dismutase (SOD), peroxidase (POD), and catalase (CAT) activity, as reported about the behavior of three different tomato genotypes [56]. These studies suggest that there are already several genes in tomatoes that make the plant tolerant to stress. These are mainly involved in the functioning of the photosynthetic apparatus and in the increase in secondary metabolites capable of protecting the plant against oxidative stress. These characteristics are due to changes that overexpress stress-related genes. However, the same characteristics have also been found in landraces [57,58]. The latter can be used in breeding programs to develop drought-resistant varieties.

To summarize, the majority of studies on the genes involved in drought tolerance in tomato plants have been undertaken through the induction of genetic mutations. Figure 1 summarizes the many stress resistance characteristics attributed to those genes. The various genes help to preserve proteins in their normal conformation, compartmentalize excess solutes, enhance water transport, keep chlorophyll intact, lower stomatal density, and scavenge ROS. In fact, as described in the next sections, some or all of these functions are already present in specific tomato varieties that are naturally more drought tolerant.

## 3. Protective Role of Proteins against Drought

Abiotic stresses affect the plant proteome, altering protein abundance, cellular localization, post-translational modifications (PTMs), protein–protein interactions, and, ultimately, the biological function of proteins [59,60,61,62]. Herein, we will focus only on those proteins that have been studied in the context of tomato biodiversity. The available literature suggests that protein profiling is indeed useful for discriminating between drought-resistant and drought-susceptible tomato varieties. Zhou et al. [63] identified a large number of proteins that were both repressed and induced in specific varieties. The proteins belong to different functional groups, suggesting a strong involvement of the protein machinery in drought stress tolerance. For example, prefoldin, which promotes protein folding without the use of adenosine-5-triphosphate (ATP), several hydrophilic proteins, and calmodulin in the calcium signal transduction pathway were found to accumulate in resistant varieties, otherwise those protein changes were not found in the susceptible tomato. A number of studies have suggested that a simple electrophoretic separation of proteins may be useful in distinguishing between tolerant and susceptible genotypes of tomato [64,65]. Therefore, it is not surprising that proteomic analysis of tomato plants identified a number of drought-responsive proteins that are primarily involved in oxidative stress response and redox status regulation. In [60], the response of four tomato genotypes to drought stress was investigated by measuring their antioxidant enzyme activities and protein expression; it was found that one genotype (EC-317-6-1) was more tolerant to drought based on the number and type of proteins that changed under stress, such as SOD, ascorbate peroxidase (APX), and CAT. These proteins could be used as markers for drought tolerance in the breeding of tomato varieties. In fact, it is well-known that drought stress increases ROS production. This affects redox homeostasis, causes oxidative stress, and reduces photosynthetic efficiency [66]. Changes in ROS metabolism and antioxidant mechanisms have also been studied in two tomato cultivars that were exposed to drought, heat, and drought plus heat (combined stress). There was a dramatic increase in SOD and APX activities in both cultivars, whereas CAT activity decreased significantly [67].

In [68], further evidence was provided for the use of antioxidant enzymes as markers. Herein, the authors evaluated the effects of three levels of drought stress on quality traits, such as antioxidant enzymes, chlorophyll content, proline content, and membrane stability, of seven tomato lines. All the traits evaluated were significantly affected by drought stress. The authors identified four genotypes to be the most tolerant and three genotypes to be the most susceptible to drought stress. Another study compared two tomato varieties (X5671R and 5MX12956) and how they coped with drought stress. Protein content and antioxidant enzyme activities were measured [69]. The authors found that X5671R was more tolerant than 5MX12956. This was because it had higher antioxidant enzyme activities. They also observed different patterns of POX, APX, and SOD isoenzymes in the two cultivars. This may indicate different roles in the stress response. The antioxidant responses of traditional tomato landraces and an industrial genotype under drought stress were also compared by [70] through the measurement of physiological, biochemical, and molecular parameters. The landraces showed higher ascorbate peroxidase and catalase activities than the industrial genotype. This was mainly due to a basal activation of this system. Alternatively, superoxide dismutase may be activated more rapidly and to a greater extent in more tolerant tomato genotypes [71]. Similarly, in a study of the effect of drought on antioxidant enzymes in native and exotic tomato genotypes [72], most of the biochemical parameters analyzed were improved in the tolerant varieties. When antioxidant enzymes were not studied directly, their effects on preventing lipid peroxidation strongly suggested their involvement in protecting different tomato genotypes from drought stress [72].

By accumulating osmolytes and hydrophilic proteins (e.g., LEA—late embryogenesis abundant proteins), plants can adjust their osmotic potential under drought conditions. Drought also causes an imbalance between electron transport and carbon assimilation during photosynthesis, which increases ROS production. Plants respond by inducing ROS-scavenging enzymes such as thioredoxin (Trx) isoforms [73]. Drought also has an effect on photosynthesis-related proteins, such as the large subunit of ribulose 1,5-bisphosphate carboxylase/oxygenase (RuBisCO) and fructose bisphosphate (FBP) aldolase [74]. Drought tolerance is also associated with an increase in stress-related proteins. These include heat shock proteins (HSPs) and late embryogenesis abundant (LEA) proteins [75].

HSPs are drought-responsive proteins that act as molecular chaperones, assisting in protein synthesis, folding, targeting, assembly, translocation, and degradation [75]. HSPs with a molecular weight of 70 kDa (HSP70) are associated with improved resistance to heat and drought [8,76,77]. Similarly, HSP70 levels increase in drought-stressed tomato plants of several cultivars [77]. The difference between four tomato cultivars (drought resistant and/or susceptible) was remarkable, demonstrating how different genotypes can have different responses in terms of chaperone proteins [77]. Since drought-tolerant tomato cultivars respond to drought stress by increasing HSP70 levels, this demonstrates the protective effect of HSP70 in drought and osmotic stress.

Cyclophilins (CYPs) are ubiquitous chaperone proteins that contain peptidyl-prolyl cis-trans isomerases that catalyze the cis-trans isomerization of an amide bond between a proline residue and the preceding amino acid residue [78]. Because of their catalytic activity, CYPs can accelerate the folding of various proteins in response to biotic and abiotic stresses. Conti and colleagues [77] found that CYP levels increased in drought-stressed tomato plants compared to controls. Thus, tomato cultivar tolerance was also associated with high expression of CYPs, which could accelerate the process of protein folding under stress conditions.

LEA proteins are protective proteins that are abundant in plants during late development and play an important role in plant survival under extreme environmental conditions. LEAs have been implicated in drought tolerance and general plant resistance to drought, salt, and cold stress. They are thought to act as water-holding molecules and are able to stabilize membranes and proteins [75]. Dehydrins are members of the LEA protein family II and play a role in the plant response to dehydration and abiotic stress in general [68,69]. Dehydrin levels were found to increase with the relative susceptibility of tomato plants; for example, the more tolerant cultivar Perina had lower dehydrin levels than the more susceptible cultivar Pisanello [77].

Other biochemical adaptations of plants to stress include the regulation of photosynthetic processes in chloroplasts [79]. Indeed, water deficit has a significant effect on chloroplast proteins [80,81]. The enzyme RuBisCO catalyzes an essential step in the Calvin cycle (carbon dioxide fixation) by generating organic molecules. It consists of a large subunit of 55 kDa and a small subunit of 14 kDa, the large subunit being encoded by the plastidial genome and the small subunit by the nuclear genome. The enzyme is a promising stress indicator. In terms of stress response, any critical damage to RuBisCO has consistent effects on the CO_2_ fixation step and thus on the synthesis of organic molecules [82]. The catalytic activity of RuBisCO decreases with the increasing duration and severity of drought [82]. This can also be explained by a decrease in protein content during stress [83], as RuBisCO degradation produces enzyme fragments detected by 2-D electrophoresis [84]. Similarly, [85] found that RuBisCO decreased in tomato leaves after prolonged drought stress. Reduced transcription of genes encoding RuBisCO small subunits may also occur under drought stress conditions, leading to a loss in enzyme stability [82]. Drought stress also affects the accumulation of different RuBisCO isoforms in tomato plants, probably due to post-translational changes [77]. Stress-induced changes can result in RuBisCO isoforms that are better adapted to a challenging environment, such as UV-B stress and heat stress [8,86]. As a result of stress, the profile of RuBisCO isoforms is altered. This results in a more functionally targeted isoforms that are better adapted to the new conditions [77]. Sucrose synthase (SuSy) is an important enzyme in sucrose metabolism, as it cleaves sucrose to form UDP-glucose and fructose. While fructose is used for respiration, UDP-glucose is a more conservative form of energy that can be used for intracellular metabolic activities as well as the synthesis of cell wall polysaccharides. SuSy activity is therefore important under drought stress scenarios because it conserves energy in UDP-glucose and increases hexose sugar content [87,88]. Plants under drought stress have a higher SuSy content than plants under irrigation [77].

Aquaporins contribute to the efficiency of photosynthesis and are also known as CO_2_ and water transporters. Aquaporins are divided into five types based on their structure and distribution; the plasma membrane intrinsic proteins (PIPs) are the ones mainly involved in CO_2_ and H_2_O transport. Overexpression of PIPs in Arabidopsis, rice, or tobacco leads to increased CO_2_ uptake in leaves. In addition, overexpression of PIPs improved drought stress responses in a variety of crops. For example, overexpression of a PIP1;2 gene in banana plants improved tolerance to both drought and salt stress. Overexpression of PIPs in tomato plants also increased drought tolerance [48,51].

Osmotin is a PR-5 family protein that helps plants in coping with different stressors. It not only protects plants from fungal, bacterial, and viral infections, but it also regulates plant water balance under drought, salt, and cold conditions. Evidence from tomato plants strongly suggests that osmotin plays a function in drought tolerance. A tobacco osmotin gene was transferred to tomato via Agrobacterium-mediated transformation, resulting in transgenic plants with higher relative water content, chlorophyll and proline content, and leaf growth than wild-type plants [49,89]. After salt or drought stress, osmotin gene expression and protein synthesis were increased in young tomato (cv. Rheinlands Ruhm) plants. There was no increase in osmotin mRNA by salt or water stress in tomato ABA-deficient mutants, showing that ABA regulates osmotin expression [90]. Conti et al. [77] studied the biochemical processes of plant resistance against drought in four Mediterranean tomato varieties. The most tolerant cultivars used distinct biochemical methods, such as different aquaporin and osmotin accumulations.

Sometimes proteins that are not normally involved in making plants drought tolerant can show peculiar patterns of behavior. This is the case with chitinase, a protein involved in the breakdown of fungal chitin, whose gene expression and protein levels were increased in more tolerant tomato genotypes [91]. Although the relationship between chitinase and drought tolerance is not known, this protein may be a useful screening tool. As chitinase has also been reported as a food allergen [92], increasing tolerance to drought stress conditions in tomato plants may increase the allergenic potential and thus undesirable side effects for consumers, a possibility that should not be overlooked. This can be much riskier if the allergenic proteins accumulate in the edible part (the fruit). Data on these points are still scarce. Tomatoes are a source of many beneficial nutrients, but some consumers must avoid tomatoes in their regular diet because of the risk of allergic reactions after consumption. Tomato allergy is immunoglobulin E (IgE)-mediated and is partly caused by pollen cross-reactivity. To date, 34 possible tomato allergens have been identified in the Allergome databases (www.allergome.org/script/search_step2.php, accessed on 16 March 2023). Allergenic proteins are involved in a wide range of biological processes, including plant growth and development, seed maturation and germination, fruit ripening, cuticle production, suberin biosynthesis, pollen development, pollen tube adhesion and growth, and defense signaling [93].

In conclusion (Figure 2), the tomato proteome is affected by drought conditions by altering protein abundance, cellular localization of proteins, PTMs, protein–protein interactions, and finally, the biological function of proteins. Most of the tomato proteins identified as drought-responsive proteins are related to different functions, such as synthesis and processing, photosynthesis, and energy production; in addition, proteins were mainly involved in oxidative stress response and redox status regulation.

## 4. Diverse Physiological Responses to Drought

Measurements of physiological activity in tomato plants exposed to drought conditions are becoming increasingly useful in distinguishing between tolerant and susceptible varieties. This is mainly because physiological analyses generally involve different sets of measurements and may link biochemical responses to whole-plant functionality. One of the key components of the photosynthetic apparatus is the PSII-LHCII supercomplex, which consists of the photosystem II (PSII) core complex and the light-harvesting complex II (LHCII). The PSII-LHCII supercomplex captures light energy and transfers it to the reaction center of PSII. Although not specific to tomato plants, drought stress can affect the structure and function of the PSII-LHCII supercomplex and induce non-photochemical quenching (NPQ) of excess light energy to prevent photodamage. Several effects at the PSII-LHCII protein level have been described. Specifically, drought reduced the photochemical efficiency of PSII and PSI and degraded light-harvesting complexes and core proteins in pea (*Pisum sativum*) leaves. The changes were probably related to the generation of reactive oxygen species [94]. Under drought stress, bundle sheath chloroplasts also exhibited higher NPQ than mesophyll chloroplasts, which was associated with dephosphorylation of LHCII subunits and increased content of PSII subunit S protein [95]. In rice, phosphorylation of PSII and LHCII proteins, together with heat dissipation, maintains photo balance [96]. In the drought-tolerant plant *Jatropha curcas*, adequate levels of photosynthetic pigments are maintained until water is available again. In addition, the quantum yields of both PSII and PSI are partially downregulated during drought, thereby protecting the photosynthetic machinery from photodamage [97]. Studies of photosynthesis in maize and sorghum showed that drought reduced photochemical quenching, the ratio of photochemical to non-photochemical processes, the effective quantum yield of photochemical energy conversion in PSII, and the rate of electron transport [98].

Tolerant and susceptible cultivars (five genotypes from an interspecific cross between *Solanum pennellii* and *S. lycopersicum*, two susceptible [UFU-22 (pre-commercial line) and cultivar Santa Clara], and one resistant [*S. pennellii*]) were found to have different water retention under controlled experimental conditions of water deficit [99]. Increased photosynthetic rates, altered stomatal closure, and decreased transpiration are other physiological variables that change in response to drought stress, resulting in improved water use efficiency. For example, Bsoul and collaborators [100] found that changes in the stomatal aperture interval were essential for increasing plant tolerance in three tomato lines, including the landrace Irhaba and the commercial cultivars Amani and GS-12.

The preliminary study of a small number of Tuscan tomato genotypes in experimental activities provided physiological data, including stomatal response, on the behavior of individual genotypes. The results showed that the Pisanello genotype was the least tolerant, while the Fragola genotype was classified as the most tolerant. The differences included biochemical indicators, such as antioxidant content, and physiological indicators [41]. The study showed that despite adaptation to the Tuscan environment and cultivation in growth chambers, tomato plants behaved differently in terms of physiological parameters. Different genotypes can be distinguished based on their unique physiological responses. When working with more genotypes, additional analyses can be added to or can replace the standard physiological measurements. For example, physio-molecular analysis was partially substituted by other markers of tomato plant health in the investigation of four commercial tomato cultivars (Imperial, Pakmore VF, Strain-B, and Tnshet Star), a drought-tolerant breeding line (L 03306), and their combinations [101]. The best way to improve the comparison of tomato genotypes is to screen additional genotypes and combine all the data using statistical analysis. PCA is a tool used to statistically evaluate large amounts of data, combining biochemical and physiological observations to categorize tomato genotypes. In the study by Aghaie [102], 14 tomato genotypes were divided into 4 categories based on different biochemical and physiological traits (antioxidants, proline, malondialdehyde, and electrolyte leakage), while in the study by Conti and colleagues [103], a set of physio-morphological traits allowed the classification of nine local Tuscan cultivars into three groups according to stress tolerance. As more data become available, information for categorizing tomato genotypes based on tolerance will become more useful for breeding and variety selection programs. Even without PCA or other statistical techniques, physiological analysis of a large number of tomato cultivars can yield a significant amount of useful information [104].

Two Mediterranean tomato landraces, Locale di Salina and Pizzutello di Sciacca, showed similar physiological responses to drought stress but differed at the biochemical and molecular levels [90]. This suggests that identifying the physiological response is sometimes easier than identifying the biochemical mechanisms. It is also an indication that the diversification of tomato landraces cannot be based on physiological data alone. In fact, the two landraces differed significantly in terms of ABA content and gene expression. The two landraces, Ciettaicale and Moneymaker, also differed significantly in ABA content and gene expression, suggesting that the criteria used to classify tomato cultivars must be carefully chosen. Evidence suggests that tomato plants differ biochemically rather than physiologically, with susceptible varieties focusing more on the accumulation of carbon-based compounds and the mobilization of starch reserves [105]. As a result, tomato genotypes may use different strategies, for example, one involving more efficient photosynthetic activity at the leaf level and the other involving carbon accumulation at the root/leaf level. The level of antioxidant response is a critical factor when analysis of physiological traits cannot distinguish or adequately explain the drought tolerance of cultivars. Comparisons between a wild (tolerant) tomato genotype and a commercial (susceptible) tomato genotype often reveal important differences at multiple levels, including pigment content, photosystem activity, and biomass production [106]. Drought stress can also lead to photorespiration, which affects plant biomass and increases ammonium availability, making tomato genotypes better able to withstand drought [107]. The importance of biochemical studies as a supporting source for physiological studies is highlighted by field studies that have investigated different tomato genotypes at the physiological, biochemical, and genetic levels. The comparison of native tomato cultivars from the Mediterranean region with more widely available cultivars revealed more significant differences. The higher basal level of ROS activity in adapted varieties, compared to susceptible ones, supported the higher tolerance of indigenous tomato varieties compared to industrial ones (Red Setter) [70]. These findings highlight the importance of physiological analysis to understand plant behavior, and the need for biochemical analysis to understand drought responses.

Different environmental stresses do not act independently, but more often plants are affected by different types of stressors. For example, heat stress and drought can both have significant effects on the physiology of tomato plants and both typically occur more or less simultaneously. Understanding how plants respond to multiple stresses is difficult, so it is important to evaluate the combined effects of stresses to determine whether plants are more susceptible to damage from a single stress, a combination of stresses, or a particular order of stresses. According to the findings on the “Roma-VF” variety, adequately hydrated plants can alleviate the detrimental effects of heat stress [108]. Water scarcity is likely the most severe physiological stress; hence, it is critical to adequately water plants under heat stress. Drought stress is likely to be the most severe stress when combined with heat stress conditions. Indeed, different tomato cultivars can be distinguished when they are stressed by drought and drought plus heat, but their responses are very similar when stressed by heat alone. This is not a general rule, as in some cases tomato cultivars can be distinguished by their response to heat stress, but if they are stressed by drought or drought plus heat, they will respond similarly. This suggests that the severity of drought stress masks the differential response to heat stress [44]. In addition, the effects of heat stress are only visible when water is a limiting factor [109]. When analyzing plants subjected to multiple stresses, additional information is needed. Fv/Fm and chlorophyll content are physiological candidates as markers of tolerance to a single stress (drought) or a combination of stresses (drought plus heat). However, other authors have suggested that ROS content may be a suitable candidate for discriminating between tolerant and susceptible cultivars to combined heat and drought stress [67]. Three different tomato cultivars (Hybrid 61, Moskvich, and Nagcarlang) were analyzed and found to be tolerant to both heat and drought stress [109]. Analyses of pigment content and ROS levels are very common because they are easy to perform as well as fast and reproducible; however, sometimes more accurate analyses are needed to differentiate between the responses of tolerant and susceptible cultivars to combined stresses. For example, multiple fluorescence excitation is a rapid and accurate genotype screening technique to assess the water status of tomato plants under drought and heat stress [110]. Lipid peroxidation, as observed in the stress-resistant cultivar Zarina [111], could be another indicator of improved physiological responses in drought-tolerant and heat-stressed genotypes. Studying how different stress conditions affect individual genotypes may reveal hidden tolerance mechanisms. For example, by exposing two uncharacterized tomato cultivars (Sufen No. 14 and Jinlingmeiyu) to a combination of stresses, novel types of resistance were identified that can be used to select genotypes with improved tolerance [67]. Although it cannot be used to distinguish between the effects of drought and heat stress on tomato plants, we feel it is important to mention that heat stress and drought stress are the two most common conditions used to identify tolerant genotypes for use in breeding programs or rootstocks.

Grafts and rootstocks are a common practice in agriculture to combine two compatible genotypes for improved tolerance to abiotic stresses. The drought-tolerant commercial hybrid “de Ramellet” genotype was grafted onto a traditional de Ramellet, with a commercial Maxifort tomato rootstock as a control [112]; this resulted in increased yield and production, as well as improved physiological performance. Two cultivars, “Strain-B” and “Super Marmande” (for which there is no evidence of drought tolerance), were grafted onto Strain-B hybrid, *Solanum pimpinellifolium* L., Edkawy cultivars, or *Datura stramonium* rootstocks, resulting in increased yield and production as well as improved physiological performance. This suggests that physiological traits may be useful markers for selecting drought-tolerant genotypes for use as rootstocks [113]. Productivity and yield can be increased in grafted plants that have received agricultural treatments, such as biochar or inoculation with mycorrhizal fungi. This requires appropriate agricultural practices, such as irrigation regimes, to maximize the value of the selected tomato genotypes. Growing potentially drought- and heat-tolerant plants in water-limited environments with high CO_2_ concentrations has shown a positive effect on plant biomass [114].

When trying to summarize the main physiological responses or adaptations of tolerant tomato cultivars to harsh environments, finding a common line is not an easy task (Figure 3). Although this may seem like an innate challenge, by contrast, it represents the potential of tomatoes to withstand drought conditions. Figure 3 summarizes the main findings known to differentiate susceptible from tolerant cultivars; it is likely that plants respond differently using a combination of mechanisms ranging from changes in tissue structures to regulated stomatal conductance and then reduced gas exchange and increased water retention, from accumulation of carbohydrates to the adaptation of a photosynthetic mechanism and an induction of photorespiration to altered accumulation of carbohydrates. Furthermore, the importance of increased abscisic acid (ABA) production should not be underestimated.

## 5. Drought Fortifies Tomato Fruit Quality

The analysis of fruit quality and quantity can be an important marker for the selection of the most suitable genotypes. Drought-tolerant plants accumulate more bioactives in fruits than susceptible cultivars under both well-irrigated conditions and drought stress, as demonstrated by the treatment of two Italian long-storage tomato landraces [115]. Selection of tolerant genotypes based on biochemical, physiological, and genetic markers, as described below, may be beneficial for improving fruit quality and water use. By the way, a study of 13 cultivars subjected to water deprivation revealed that bioactives accumulated more in the peel than in the pulp [116].

Tomato fruit is known for its high concentration of health-promoting biomolecules such as carotenoids, vitamin C, and hydroxycinnamic acids, all of which are beneficial for human health [117,118]. A pertinent question, however, is how drought stress affects the levels of these nutraceutical compounds in tomato fruit. The answer is not simple, because the effects of drought stress on nutraceutical compounds vary depending on both the severity of the stress and the compound in question. Nevertheless, it is known that different stressors, including drought, can induce a significant increase in secondary metabolites as a defense mechanism against the production of reactive oxygen species (ROS) or free radicals [119,120], which, if they exceed certain thresholds, could become extremely harmful. They are known to cause oxidative damage to essential cellular components such, as proteins, DNA, and lipids. Since biodiversity is a resource of wild or locally adapted cultivars that are both differentially rich in biomolecules and differentially stress tolerant, the choice of the specific cultivar under study is an additional factor that can influence our understanding of the response to drought stress. Numerous studies have been conducted to investigate the characteristics of different tomato cultivars to better understand their different behaviors. For example, drought stress has been shown to affect the shikimate pathway and phenolic compounds in more susceptible cultivars. Conversely, the more tolerant cultivar “Zarina” increases the activity of flavonoids and certain components of the shikimate pathway [107]. The effect of drought stress on the content of bioactive compounds in fruits varies among cultivars. Several studies in the literature have investigated this phenomenon by comparing different genotypes. For example, under drought stress conditions, ‘Matina’, a cultivar of German origin reported to be drought tolerant, was found to have an increase in both vitamin C and lycopene content [121]. In a similar study, Guida et al. [115] found that ‘Locale di Salina’ and ‘Piennolo del Vesuvio’ cultivars had higher levels of bioactive compounds under similar drought stress conditions compared to the same cultivars in a fully irrigated regime. Furthermore, in a recent study by Conti and colleagues [116], “Quarantino”, which showed good resistance to drought, was found to have higher levels of bioactive compounds than other local and commercial cultivars under drought stress. Additional studies have found an increase in total polyphenol content (TPC) in specific cultivars under drought stress conditions. For example, the Ethiopian cultivar “Cochoro”, which has been shown to perform well under deficit irrigation, showed an increase in TPC under such conditions [121]. Similarly, the drought-tolerant cultivar ‘Perina’ was found to have higher levels of antioxidants, flavonoids, and TPC compared to other local and commercial cultivars [116]. Furthermore, when exposed to drought stress, three local Sicilian landraces increased their polyphenol content; even after rehydration, they maintained higher levels of bioactive compounds than fully irrigated cultivars, highlighting the importance of rehydration after prolonged drought periods to maintain good levels of yield and bioactive compounds in the fruit [122]. The challenge of maintaining high yields and fruit quality under harsh environmental conditions, such as drought is a pressing concern for future agriculture. Water scarcity typically leads to a significant reduction in productivity. However, recent research has shown that fruit quality can be improved under drought conditions. For example, four *Solanum pennellii* introgression lines, including two drought-resistant and two drought-susceptible lines, showed a 66% decrease in yield but an increase in fruit quality when water was scarce [123]. The question is whether it is possible to maintain yield while improving quality under drought. The use of drought-tolerant landraces could be an effective strategy to conserve water, allowing a rational use of water resources without significantly reducing yield. Drought tolerance has been demonstrated in studies of the Italian long-storage tomato landraces ‘Locale di Salina’ and ‘Piennolo del Vesuvio’, with slightly lower yields compared to fully irrigated plants [115]. Fullana-Pericàs and colleagues [57] also studied 165 tomato genotypes under water deficit irrigation and found several landraces with promising drought tolerance. Different genotypes are affected to a different extent by drought, so it is possible to observe a significantly smaller reduction in fruit yield for LE 118, LE 58, or LE 114, but not for LE 1 and LE 125, which gave poor yields [124]. It can be concluded (Figure 4) that agriculture would benefit from biodiversity and the potential implementation of drought-stress tolerant genotypes, which would make agriculture more sustainable with less water usage while maintaining high fruit quality.

## 6. Conclusions

Biodiversity is thought to be an important genetic resource for numerous aspects of plant physiology and development, although the extent of its impact remains uncertain. It is believed to play a critical role in maintaining ecosystem balance and potentially provides various ecosystem services, such as air and water purification, nutrient recycling, and climate regulation. The tomato, a widely cultivated annual crop with significant economic value worldwide, is known to grow in regions characterized by high temperatures and challenging growing conditions. Tomato breeding programs, which draw upon the genetic diversity found in wild tomato species, have been developed to enhance the plant’s adaptability to such adverse conditions. However, the effectiveness and success of these programs are not fully established, leaving gaps in our understanding. Conservation efforts aimed at preserving tomato genetic resources are considered vital to ensure the long-term sustainability of tomato production and food security, although the specific implications and outcomes are uncertain. Consequently, it is postulated that the characterization of the extensive tomato genetic resources is of utmost importance for the advancement of more tolerant cultivars. It is suggested that traits associated with drought tolerance may potentially exist at multiple levels, ranging from genetic factors to proteins to physiological mechanisms, as suggested in various chapters of this review. The notion is based on the observation that a substantial number of manuscripts in the literature indicate that the increased tolerance of tomato plants to harsh conditions is unlikely to be attributable to a singular parameter, but rather a combination of several factors. However, the precise nature of these critical traits and the interconnectedness between different parameters remain elusive.

Emerging evidence hints at the potential importance of specific genes in enhancing drought tolerance, alongside the indispensability of physiological adaptations for coping with extreme environmental conditions. Yet, further investigations are needed to elucidate the intricate relationships between these responses at various levels. It is hypothesized that alterations in the expression of one or more genes might be influential in boosting the drought tolerance of tomato plants, but it is crucial that they are accompanied by suitable biochemical and physiological changes. In practice, this supports the concept that drought tolerance is a complex and multifaceted phenomenon, though the exact mechanisms are not well-defined.

This understanding represents a significant milestone as it can serve as the foundation for improving tomato varieties through traditional breeding programs, as well as utilizing genetic selection methods, such as marker-assisted selection or cutting-edge genetic approaches, such as Crispr-Cas9. By gaining insights into the intricate relationships between different parameters implicated in drought tolerance, it is hoped that more effective strategies for enhancing tomato cultivars can be devised. Nevertheless, the practical implications and success of such strategies remain uncertain. Moreover, it is worth noting that a comprehensive comprehension of the genetic, biochemical, and physiological relationships among different tomato cultivars can have a profound impact on the development of new cultivars, especially in countries where genetic transformation is not allowed, such as Italy. These findings could potentially enhance breeding possibilities and facilitate the generation of more well-suited plants. Countries, such as Italy have invested significant effort in safeguarding and promoting tomato varieties and cultivars, aiming to preserve their genetic background for future utilization in anticipation of climate change and to enhance economic competitiveness. Additionally, it is crucial to consider that any selection for drought tolerance should not compromise the quality of the fruit. For instance, minimizing allergenic compounds or their absence may be desired, while simultaneously promoting and improving the nutritional value of the tomato fruit by enriching it with natural products beneficial to human health. However, the practical implementation and realization of these goals remain uncertain and require further investigation.

## Figures and Tables

**Figure 1 ijms-24-10044-f001:**
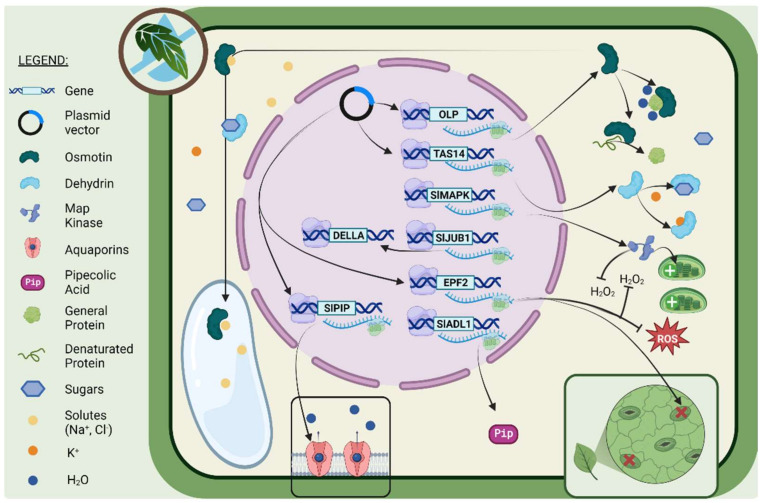
Scheme of the main gene activities that allow some tomato cultivars to be drought tolerant. Among the main changes in gene expression, the diagram shows those that are better characterized, such as OLP, TAS14, SlMAPK, SlJUB1, EPF2, DELLA, and SlPIP, and the function of the proteins encoded by these genes. For more information, please refer to the text. Image was created with the tools of BioRender (https://app.biorender.com/ (accessed on 5 June 2023)).

**Figure 2 ijms-24-10044-f002:**
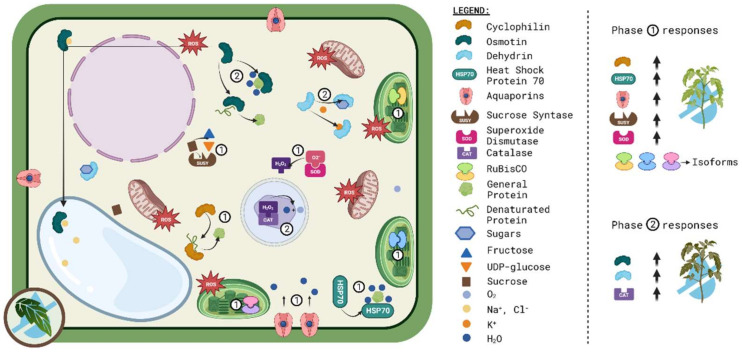
Summary of the main biochemical mechanisms used by tomato plants to cope with drought. In general, the protein-based mechanisms can be divided into two groups according to the time of response. Phase 1 response involves changes in cyclophilin, HSP70, aquaporins, SuSy, and SOD, in addition to changes in RuBisCO isoforms. Phase 2 includes changes in osmotin, dehydrin, and catalase content and activity. Image was created with the tools of BioRender (https://app.biorender.com/ (accessed on 30 May 2023)).

**Figure 3 ijms-24-10044-f003:**
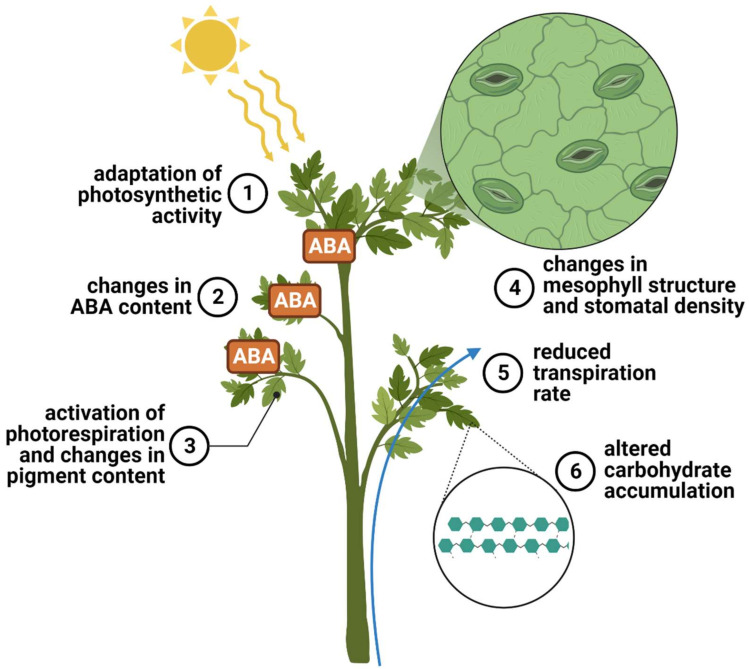
Summary of the main physiological mechanisms that allow tomato plants to escape drought. Mechanisms are those that are reported to vary between tomato cultivars and that may allow one cultivar to be more tolerant than another. Image was created with the tools of BioRender (https://app.biorender.com/ (accessed on 28 May 2023)).

**Figure 4 ijms-24-10044-f004:**
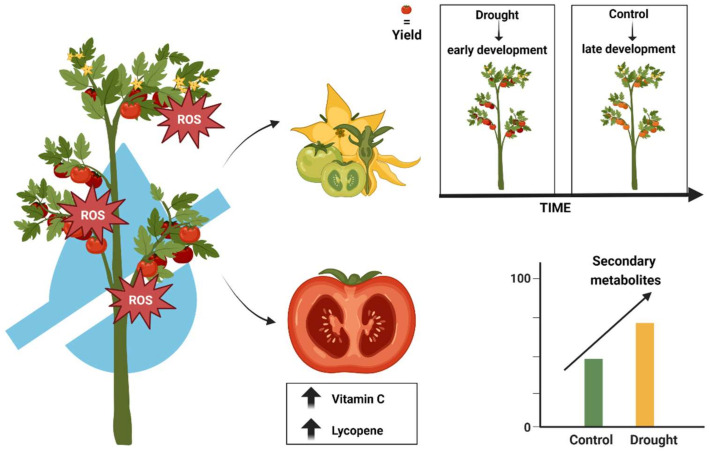
Summary of the effects of drought stress on tomato fruit quality. ROS production following drought stress is known to induce early fruit development compared to irrigated plants, although yield is not affected. In addition, drought stress is reported to increase the content of bioactive molecules, such as vitamin C and lycopene. Image was created with the tools of BioRender (https://app.biorender.com/ (accessed on 5 June 2023)).

## Data Availability

The data presented in this study are available on request from the corresponding author.

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
