# Peer review of "Tomato Biodiversity and Drought Tolerance: A Multilevel Review"

_ijms, 2023, doi:10.3390/ijms241210044_

Round 1

Reviewer 1 Report

In this manuscript authors aim to summarize and emphasize the importance of tomato biodiversity in response to drought at the physiological, genetic, biochemical, and fruit quality levels. Authors claimed that the response of plants to drought stress is complex and involves coordination between gene expression, biochemical responses, physiological adaptation, and their integration. In tomato (Solanum lycopersicum L.), drought limits the productivity of the plant and thus the economics of global production. Authors also claimed that  many different tomato cultivars are currently catalogued and cultivated; they differ in terms of genetic, biochemical and physiological traits, thus representing a reservoir of potential candidates for coping with drought stress. Authors also claimed that some reports have compared different tomato cultivars in response to drought conditions, but a comprehensive review of the contribution of specific physiomolecular traits and how they vary among cultivars is not available.

Question 1: The aims of the manuscript and the results of the data clearly and concisely stated in the abstract? Please revise accordingly.

Question 2: The authors must discuss importance of the study and innovation of this study because there are many articles on this topic. This will make the article more attractive for the readers of IJMS.

Question 3: Under the paragraph ‘gene base resistance to drought’ authors should add a table showing list of genes identified for tolerance to drought stress in tomato, along with strategy used and reference.

Question 4: Some of the sentences are long and not cited. Please cite every statement properly in the manuscript.

Question 5: Introduction section is well written, sentences and paragraphs coherence are balanced. But there are some grammatical and typo mistakes which must be revised.  Introduction must also discuss molecular studies and approaches for drought resistance and management by different approaches, crop productivity and stress response. Authors should also cite the following recent literature doi.org/10.1007/s10725-021-00785-7, doi.org/10.1016/j.jplph.2019.152997 and doi.org/10.1007/s00427-019-00643-7

Question 6: There is only one Figure in this review article. Authors should also draw a figure showing the molecular mechanism of drought stress in tomato.

Question 7: Authors should draw a table showing physiological and molecular strategies and traits of tolerance in different plant species against the drought stress. Without proper figures/tables your manuscript will look tired some for the readers and authors will not be able to pick a quick idea.

Question 8: A sentence depicting the conclusion of the paragraphs should be added to summarize your findings at the end of ever paragraph.

Question 9: Moreover, the author must add future perspectives of this study in the introduction section.

Question 10: Authors should clearly state in the conclusion that what they have identified in their research?

Minor English corrections required

Author Response

REVIEWER 1
In this manuscript authors aim to summarize and emphasize the importance of tomato biodiversity in response to drought at the physiological, genetic, biochemical, and fruit quality levels. Authors claimed that the response of plants to drought stress is complex and involves coordination between gene expression, biochemical responses, physiological adaptation, and their integration. In tomato (Solanum lycopersicum L.), drought limits the productivity of the plant and thus the economics of global production. Authors also claimed that many different tomato cultivars are currently catalogued and cultivated; they differ in terms of genetic, biochemical and physiological traits, thus representing a reservoir of potential candidates for coping with drought stress. Authors also claimed that some reports have compared different tomato cultivars in response to drought conditions, but a comprehensive review of the contribution of specific physio-molecular traits and how they vary among cultivars is not available.
In the revised version of the manuscript, according to reviewer criticisms, we made several modifications, and the modified text has been highlighted in yellow.
Question 1: The aims of the manuscript and the results of the data clearly and concisely stated in the abstract? Please revise accordingly.
Answer: First of all, we would like to thank the reviewer for taking the time to review our Ms. We appreciate your critical review and constructive suggestions for improvement. We agreed that the abstract should report more information, and it was changed accordingly.
Question 2: The authors must discuss importance of the study and innovation of this study because there are many articles on this topic. This will make the article more attractive for the readers of IJMS.
Answer: We thank the reviewer for the opportunity to specify the importance of this study. Indeed, the widespread cultivation of tomato has generated many articles and reviews on the relationship between tomato and drought. However, we believe that the evaluation of the relative contribution of different tomato cultivars and their multilevel study is a distinctive feature of this manuscript. We have added an additional paragraph at the end of the Introduction that emphasises this concept.
Question 3: Under the paragraph ‘gene base resistance to drought’ authors should add a table showing list of genes identified for tolerance to drought stress in tomato, along with strategy used and reference.
Answer: We thank the reviewer for the suggestion. However, when we were thinking about a table, we also had in mind the possibility of adding multiple schemes. We found that tables and schemes should repeat the same concept in some way. So in our hand, schemas have more impact and can convey a message very quickly to the reader. So we decided to add more schemes, but not the table. We hope the reviewer can appreciate our choice.
Question 4: Some of the sentences are long and not cited. Please cite every statement properly in the manuscript.
Answer: We thank the reviewer for his/her critical reading of our MS. The entire MS has been revised and several sentences have been changed and rewritten. In addition, we have cited each statement that we have not previously cited.
Question 5: Introduction section is well written, sentences and paragraphs coherence are balanced. But there are some grammatical and typo mistakes which must be revised. Introduction must also discuss molecular studies and approaches for drought resistance and management by different approaches, crop productivity and stress response. Authors should also cite the following recent literature doi.org/10.1007/s10725-021-00785-7, doi.org/10.1016/j.jplph.2019.152997 and doi.org/10.1007/s00427-019-00643-7
Answer: Thanks to the reviewer for his/her criticism and suggestions. We revised the entire manuscript for grammatical errors. In addition, we added a couple of paragraphs in the Introduction about molecular approaches used to analyse drought tolerance. We also cited two of the suggested papers.
Question 6: There is only one Figure in this review article. Authors should also draw a figure showing the molecular mechanism of drought stress in tomato.
Answer: Thank you for making this observation. We realised that using only one figure can be restrictive and does not provide a complete picture of the drought-resistance mechanisms used by tomato plants. As a result, we decided to add more schemes, one for each main section, and to remove the last figure.
Question 7: Authors should draw a table showing physiological and molecular strategies and traits of tolerance in different plant species against the drought stress. Without proper figures/tables your manuscript will look tired some for the readers and authors will not be able to pick a quick idea.
Answer: We thank the reviewer for the observation. In fact, a visual representation of the main findings may help readers. We decided to present the main mechanisms that tomato plants implement against drought by adding more schemes. Since the use of tables and schemes could lead to repetition of results and make the paper heavy, we decided to use only figures.
Question 8: A sentence depicting the conclusion of the paragraphs should be added to summarize your findings at the end of ever paragraph.
Answer: The reviewer's helpful criticism is appreciated. At the end of each paragraph, a summary sentence was added.
Question 9: Moreover, the author must add future perspectives of this study in the introduction section.
Answer: We thank the reviewer for this comment. However, we believe that the Conclusion paragraph is a better place to describe the potential of the study and how it can contribute to improving the tolerance of tomato cultivars in anticipation of climate change. We have modified the Conclusion section by adding more information and hypotheses, but we prefer to keep the future perspectives in this section and not move them to the Introduction.
Question 10: Authors should clearly state in the conclusion that what they have identified in their research?
Answer: Thanks to the reviewer for useful comments. The conclusion part was revised adding more specific information about our review’s aim and scope.

Reviewer 2 Report

In thins manuscript, the authors reviewed the role of tomato diversity under drought stress. They provide all aspects of tomato reactions including the physiological, genetic, biochemical, and fruit quality in response to drought.  

In my opinion, the current version has good potential for publishing in IJMS.

Author Response

REVIEWER 2

In this manuscript, the authors reviewed the role of tomato diversity under drought stress. They provide all aspects of tomato reactions including the physiological, genetic, biochemical, and fruit quality in response to drought. In my opinion, the current version has good potential for publishing in IJMS.

Answer: Thanks to the reviewer for taking the time to review our MS and for the positive comment.

Reviewer 3 Report

The manuscript «Tomato Biodiversity and Drought Tolerance: A Multilevel Review» by Veronica Conti, Luigi Parrotta, Marco Romi, Stefano Del Duca and Giampero Cai is devoted to the observation of  protective mechanisms in tomato plants to drought stress.

 After rather long Introduction section, which is written in rather popular science style and containing the most general information about the ecological state of the planet, as well as about Solanum lycopersicum L. as a plant species, the authors move on to the “Gene-based resistance to drought” section. The genetic part itself starts from “Although there are other reviews on this topic [27, 28], we will only present some specific examples of how genes affect the behavior of tomato plants to make them more drought tolerant.” Since this manuscript is being considered for publication in the IJMS, it is necessary to make a detailed review of the currently available literature on the genetic mechanisms of drought tolerance, with an emphasis on what is currently known about S. lycopersicum. In addition, it is necessary to start by characterizing the stress tolerance of S. lycopersicum in different cultivars. These sections are unexpectedly found in the second part of the review. These are “Diverse physiological responses to drought” and “Drought fortifies tomato fruit quality”. These parts are much more informative and better written. Definitely, these are the parts that should be moved to the beginning of the review.

Besides, in introduction, please, also give some references to literature data, concerning the examples of «molecular phenotyping techniques» and expand this part.

 Returning to the “Gene-based resistance to drought” section, the sentences “MdEPF2 is a functional ortholog of EPF2 in Arabidopsis and can be used to improve drought tolerance and water use efficiency (WUE) in crops. An epidermal patterning factor (EPF) from apple (Malus domestica) was ectopically expressed in S. lycopersicum cultivar 'Micro-Tom', and transgenic plants showed higher values for relative leaf water content, chlorophyll, photosynthetic rates, and WUE than wild type ( WT) [29]” are unclear. It is completely unclear what MdEPF2 in drought tolerance mechanisms is, as well as the subsequent abbreviations. This section needs significant revision and additions. It would also be highly desirable to add a drawing with schemes of the functioning of known genes encoding the mechanisms of drought tolerance. In “Protective role of proteins against drought”: “Specific proteins such as prefoldin, hydrophilic proteins, and calmodulin were found to accumulate in resistant varieties” authors should have specified in detail what the protective role of these proteins is.

In the sentence «Overexpression of the dehydrin gene (TAS14) reduces osmotic potential while increasing solute (sugar and K+) and abscisic acid (ABA) content, resulting in improved plant performance under drought stress [36]» it is difficult to understand what does «plant performance» mean. Is it connected with the performance of photosynthetic apparatus and how was it measured? In Figure 1 « photosynthetic efficiency»  was also mentioned, however it is not described well. Besides, in the part «Gene-based resistance to drought» terms «photosynthetic rates» and «photosynthetic efficiency» should be indicated. How were they measured?

In the part «Diverse physiological responses to drought» authors should give information about physiological markers of stress and how do plants adapt to heat, drought and multiple stresses. Any information about changes in the proteins, which are the compounds of PSII-LHCII supercomplex as well as changes in non-photochemical changes is missed. Such information should be mentioned, at least, briefly.

Figure 1, which is in the end of the text is rather primal. Such scheme may be suitable for an Introduction section to familiarize the reader with the problem at hand. By 2023, a significant amount of knowledge about the mechanisms of development of drought resistance has been accumulated in the world literature. The schemes of the functioning of the genetic mechanisms of drought resistance in plants in general and, probably, in tomatoes in particular should be added.

Abbreviation “IPCC” should be specified, as well as SOD, POD and CAT. The last ones were specified much later after the first mention.

Author Response

REVIEWER 3

The manuscript «Tomato Biodiversity and Drought Tolerance: A Multilevel Review» by Veronica Conti, Luigi Parrotta, Marco Romi, Stefano Del Duca and Giampero Cai is devoted to the observation of  protective mechanisms in tomato plants to drought stress.

In the revised version of the manuscript, according to reviewer criticisms, we made several modifications, and the modified text has been highlighted in yellow.

After rather long Introduction section, which is written in rather popular science style and containing the most general information about the ecological state of the planet, as well as about Solanum lycopersicum L. as a plant species, the authors move on to the “Gene-based resistance to drought” section. The genetic part itself starts from “Although there are other reviews on this topic [27, 28], we will only present some specific examples of how genes affect the behavior of tomato plants to make them more drought tolerant.” Since this manuscript is being considered for publication in the IJMS, it is necessary to make a detailed review of the currently available literature on the genetic mechanisms of drought tolerance, with an emphasis on what is currently known about S. lycopersicum. In addition, it is necessary to start by characterizing the stress tolerance of S. lycopersicum in different cultivars. These sections are unexpectedly found in the second part of the review. These are “Diverse physiological responses to drought” and “Drought fortifies tomato fruit quality”. These parts are much more informative and better written. Definitely, these are the parts that should be moved to the beginning of the review.

Answer: We thank the reviewer for this suggestion. However, we prefer to present the data on drought tolerance in a logical order. We started with gene expression and then moved to the upper levels, such as biochemical and physiological, and ended with fruit quality. To follow the reviewer's suggestion, we have moved the first paragraph of the physiology section to the top of the gene section. We have also made several changes to the gene section, reorganising the text and adding more information. We hope that the gene section is now more fluent and that the reviewer appreciates our intention to keep the gene section at the beginning of the paper.

Besides, in introduction, please, also give some references to literature data, concerning the examples of «molecular phenotyping techniques» and expand this part.

Answer: We thank the reviewer for the comment. We agreed with the reviewer about the importance of phenotyping techniques and we added a small paragraph in the introduction section about them. We decided to maintain the review focus on the biodiversity of tomatoes in response to drought stress and for this reason, we decided to limit the addition of information.

Returning to the “Gene-based resistance to drought” section, the sentences “MdEPF2 is a functional ortholog of EPF2 in Arabidopsis and can be used to improve drought tolerance and water use efficiency (WUE) in crops. An epidermal patterning factor (EPF) from apple (Malus domestica) was ectopically expressed in S. lycopersicum cultivar 'Micro-Tom', and transgenic plants showed higher values for relative leaf water content, chlorophyll, photosynthetic rates, and WUE than wild type ( WT) [29]” are unclear. It is completely unclear what MdEPF2 in drought tolerance mechanisms is, as well as the subsequent abbreviations.

Answer: Thanks to the reviewer for pointing this out. We apologise for our mistake and the acronym and specific role of MdEEPF have been added to the manuscript.

This section needs significant revision and additions. It would also be highly desirable to add a drawing with schemes of the functioning of known genes encoding the mechanisms of drought tolerance.

Answer: Thanks to the reviewer. We added a summary figure for all paragraphs.

In “Protective role of proteins against drought”: “Specific proteins such as prefoldin, hydrophilic proteins, and calmodulin were found to accumulate in resistant varieties” authors should have specified in detail what the protective role of these proteins is.

Answer: Thanks to the reviewer for the comment. We revised the sentence and we added missed information.

In the sentence «Overexpression of the dehydrin gene (TAS14) reduces osmotic potential while increasing solute (sugar and K+) and abscisic acid (ABA) content, resulting in improved plant performance under drought stress [36]» it is difficult to understand what does «plant performance» mean. Is it connected with the performance of photosynthetic apparatus and how was it measured? In Figure 1 « photosynthetic efficiency» was also mentioned, however it is not described well. Besides, in the part «Gene-based resistance to drought» terms «photosynthetic rates» and «photosynthetic efficiency» should be indicated. How were they measured?

Answer: We thank the reviewer for this constructive comment. We have added more information on plant performance. Specifically, we reported that in the cited paper, the authors reported shoot and fruit biomass as plant performance. We have also indicated what is meant by photosynthetic rate and efficiency, which in most cases is the Fv/Fm ratio.

In the part «Diverse physiological responses to drought» authors should give information about physiological markers of stress and how do plants adapt to heat, drought and multiple stresses. Any information about changes in the proteins, which are the compounds of PSII-LHCII supercomplex as well as changes in non-photochemical changes is missed. Such information should be mentioned, at least, briefly.

Answer: We thank the reviewer for this comment. We have emphasised the different contribution of drought and heat stress to tomato cultivars and provided additional information on the potential markers that can be used to discriminate between cultivars under both stress conditions.

As for the PSII-LHCII proteins, we have added a paragraph to highlight the main findings, although they are not specific to tomato plants. However, the information may be useful to evaluate photosynthesis-related changes after drought stress.

Figure 1, which is in the end of the text is rather primal. Such scheme may be suitable for an Introduction section to familiarize the reader with the problem at hand. By 2023, a significant amount of knowledge about the mechanisms of development of drought resistance has been accumulated in the world literature. The schemes of the functioning of the genetic mechanisms of drought resistance in plants in general and, probably, in tomatoes in particular should be added.

Answer: We thank the reviewer for the accurate comment and revision. In light of the similar comment of the reviewer 1, we decided to remove the final conclusive figure. We added a specific figure/scheme for each paragraph. Now, we believe that the presence of a figure at the end of every paragraph could be useful for readers.

Abbreviation “IPCC” should be specified, as well as SOD, POD and CAT. The last ones were specified much later after the first mention.

Answer: Thanks to the reviewer for the accurate revision of our MS. We specified the IPCC acronym and added the full names of the enzymes. In addition, we have reviewed the entire MS and corrected some other grammatical and typographical errors.

Round 2

Reviewer 1 Report

This study aimed to summarize the contribution of specific physio-molecular traits to drought tolerance and how they vary among tomato cultivars. However, there are some deficiencies which must be addressed.

In the abstract section it would be better to specify the cultivar names of the tomato.

Also add some main findings of this review in the abstract section. Because the abstract is only focused on the introduction.

“Numerous studies have shown that both” only one study is cited here please add more. https://doi.org/10.3390/molecules28052292, https://doi.org/10.1007/s10725-021-00785-7, https://doi.org/10.3390/ijms22179175

Line “limiting the supply of carbon dioxide (CO2)” could be cited with recent study here doi: 10.1039/d2gc02467e

Line “ chlorophyll content and increased proline concentration” cold be cited with relevant study doi: 10.1016/j.jhazmat.2022.128981

Revise the sentence replace the word want with other word “We also want to emphasize how drought response”

Last paragraph of the introduction which discuss aim of the study is confusing. Such as in the start of the paragraph “This review will focus on different molecular mechanisms” then “is a critical assessment of the importance of conserving and valorizing tomato biodiversity as a gene pool” followed by “Here, we emphasize (1) the impo…” I mean to say repetition of the aims and objectives can be found here.

I recommend to elaborate the aims and objectives one time and clear way looking at the contents of the review.

In gene based resistance to drought only few genes are presented in the scheme. It would be better to add a table showing main genes, pathways and functions and proteins.

This section is the core of the study table should be added for this section presenting responses, draught effects, areas, cultivar names, and genes.

In conclusion add gaps for the future studies.

Long and void sentences should be avoid to convey clear message to readers

Author Response

Responses to review

This study aimed to summarize the contribution of specific physio-molecular traits to drought tolerance and how they vary among tomato cultivars. However, there are some deficiencies which must be addressed.

In the abstract section it would be better to specify the cultivar names of the tomato.

Response: Thank you for your feedback. We discussed this issue and decided that it is preferable not to include cultivar names in the abstract section. We cited many cultivars in the manuscript, and if they were included in the abstract, it would appear as a list of names; furthermore, the abstract has a word limit, which we have already exceeded due to the addition of more specific findings.

Also add some main findings of this review in the abstract section. Because the abstract is only focused on the introduction.

Response: Thank you for your input. As stated in the preceding response, the abstract has been revised in response to the reviewer's suggestions, and it now contains more specific information about the main findings.

“Numerous studies have shown that both” only one study is cited here please add more. https://doi.org/10.3390/molecules28052292, https://doi.org/10.1007/s10725-021-00785-7, https://doi.org/10.3390/ijms22179175

Response: Thank you for the suggestion, but we believe the articles are not strictly relevant to our review. The first refers to heavy metal pollution, while the second refers to fruit ripening and flowering. As a result, only one of the proposed articles has been included.

Line “limiting the supply of carbon dioxide (CO2)” could be cited with recent study here doi:10.1039/d2gc02467e

Response: Thank you for the recommendation, but we believe the suggested article is not strictly relevant to our review because it refers to CO2 capture by a sulfur bacterium.

Line “ chlorophyll content and increased proline concentration” cold be cited with relevant study doi: 10.1016/j.jhazmat.2022.128981

Response: Thank you for your suggestion, but we believe that the recommended article is not strictly relevant to our review. However, in order to reinforce the sentence, we decided to include a more relevant paper.

Revise the sentence replace the word want with other word “We also want to emphasize how drought response”

Response: Thanks to the review. We replaced “want” with “attempted”.

Last paragraph of the introduction which discuss aim of the study is confusing. Such as in the start of the paragraph “This review will focus on different molecular mechanisms” then “is a critical assessment of the importance of conserving and valorizing tomato biodiversity as a gene pool” followed by “Here, we emphasize (1) the impo…” I mean to say repetition of the aims and objectives can be found here. I recommend to elaborate the aims and objectives one time and clear way looking at the contents of the review.

Response: Thank you for your advice. The final paragraph of the introduction did sound repetitive. For this reason, we rewrote the paragraph, eliminating recurrent sentences and streamlining the text.

In gene based resistance to drought only few genes are presented in the scheme. It would be better to add a table showing main genes, pathways and functions and proteins.

This section is the core of the study table should be added for this section presenting responses, draught effects, areas, cultivar names, and genes.

Response: We appreciate the reviewer's feedback, but we believe that the figure on gene expression is more helpful than the table. The reason for this is that the figure has a considerably greater influence on the reader than the table. Furthermore, most genetic pathways do not result in protein production since several genes critical for drought tolerance code for transcription factors whose activity is currently unknown. As a result, we opt to keep the figure and avoid adding a table. We believe that the image serves a dual purpose in that it both summarizes the majority of the information and invites readers to read the content.

In conclusion add gaps for the future studies.

Response: Thank you very much. The section "Conclusion" has been updated. As you correctly pointed out, there are still some unknown issues and gaps that need to be filled. We have now changed the text to emphasize what is unknown and thus what needs to be done in the future.

Long and void sentences should be avoid to convey clear message to readers

Response: Thank you for your advice. The entire manuscript has been revised, with some sentences rewritten or corrected. To make it easier to follow our modifications, only the major changes have been highlighted in yellow.

Reviewer 3 Report

The authors of the present manuscript have significantly improved its content. Thus, the manuscript may be accepted for publication.

Author Response

Dear Reviewer, thank you for your positive comment